# Facilitated Transport across Glycocalyceal Barriers in the Chick Chorioallantoic Membrane

**DOI:** 10.3390/polym16010004

**Published:** 2023-12-19

**Authors:** Anuhya Dayal, Jennifer M. Pan, Stacey P. Kwan, Maximilian Ackermann, Hassan A. Khalil, Steven J. Mentzer

**Affiliations:** 1Laboratory of Adaptive and Regenerative Biology, Brigham & Women’s Hospital, Harvard Medical School, Boston, MA 02115, USA; anuhya.dayal@outlook.com (A.D.); jmpan@bwh.harvard.edu (J.M.P.); spkwan@bwh.harvard.edu (S.P.K.); hakhalil@bwh.harvard.edu (H.A.K.); 2Institute of Functional and Clinical Anatomy, University Medical Center, Johannes Gutenberg University, 55131 Mainz, Germany; maximilian.ackermann@uni-mainz.de

**Keywords:** hydrogel, pectin, color tracers, planimetry

## Abstract

Targeted drug delivery to visceral organs offers the possibility of not only limiting the required dose, but also minimizing drug toxicity; however, there is no reliable method for delivering drugs to the surface of visceral organs. Here, we used six color tracers and the chick chorioallantoic membrane (CAM) model to investigate the use of the heteropolysaccharide pectin to facilitate tracer diffusion across the glycocalyceal charge barrier. The color tracers included brilliant blue, Congo red, crystal violet, indocyanine green, methylene blue, and methyl green. The direct application of the tracers to the CAM surface or embedding tracers into linear-chain nanocellulose fiber films resulted in no significant diffusion into the CAM. In contrast, when the tracers were actively loaded into branched-chain pectin films, there was significant detectable diffusion of the tracers into the CAM. The facilitated diffusion was observed in the three cationic tracers but was limited in the three anionic tracers. Diffusion appeared to be dependent on ionic charge, but independent of tracer size or molecular mass. We conclude that dye-loaded pectin films facilitated the diffusion of color tracers across the glycocalyceal charge barrier and may provide a therapeutic path for drug delivery to the surface of visceral organs.

## 1. Introduction

Conventional drug delivery is typically accomplished through oral administration [1]. Because of suboptimal targeting, oral administration often requires high doses and repeated administration. Protein and peptide drugs commonly used in biologic therapies are particularly inefficient. Protein drugs typically have short circulation times with serum half-lives of only minutes to hours [2]. Improved efficiency of drug delivery requires a method of regulating drug availability over time and space; that is, a method of directing the drug to the target organ. Targeted drug delivery offers the possibility of not only a reduced required dose, but also of minimizing drug toxicity [3].

Previous attempts to target epithelial membranes have had some success with topical applications on the skin [4], cornea [5], nasal mucosa [6], and airway epithelium [7]; however, there are no reliable methods for the targeting visceral organ epithelium (also known as mesothelium). There are several obstacles to targeting visceral organ epithelium. The first obstacle is the physical glycocalyceal barrier. Also referred to as the mesopolysaccharide (MPS), the visceral organ surface is covered with a dense and negatively charged structure recently shown to be 100- to 1000-fold thicker than previous estimates [8]. The function of the glycocalyx appears to be the maintenance of epithelial hydration and the minimization of friction between organ surfaces [9]. The slippery character of the glycocalyx presents a second obstacle to targeted delivery. Because of the glycocalyx structure, it is difficult to maintain the prolonged contact necessary for drug delivery. The glycocalyx structure appears to be a significant barrier to both passive diffusion and paracellular transport.

The bioadhesive hydrogel called pectin presents a potential solution to targeted visceral organ drug delivery. A naturally occurring structural heteropolysaccharide, pectin, is abundant in nature. Pectin comprises approximately 30% of the primary cell walls of plants. Pectin provides structural support by entangling with cellulose microfibrils and other pectin chains. The most abundant component of pectin is homogalacturonan, a glycan of α1→4-linked D-galacturonic acid that can be largely carboxy methyl esterified. Relevant to drug delivery, pectin has a significant water (free) volume available for drug loading [10]. In addition, pectin films have a branched-chain structure that physically entangle with the glycocalyx on the epithelial surface [11]. Entanglement results in strong bioadhesion and extended contact. The intimate contact of pectin films to the visceral organ surface may facilitate the targeted delivery and potential absorption across the visceral organ mesothelium.

The ex ovo chick chorioallantoic membrane (CAM) model is a useful model for evaluating the pectin hydrogel [12]. The main advantage of the CAM model is that it is an in vivo model representing a complex physiologic environment [13]. The CAM model is low cost and easily accessible. A major consideration for these experiments is that the color tracers can be directly visualized for photodocumentation. Also, the CAM has an immature immune system and undeveloped nervous system during the experimental time frame [14]. The practical result that there is little confounding inflammation and ethics committee review is not needed [15].

In this report, we used the chick chorioallantoic membrane (CAM) model to assess the functional properties of the pectin hydrogel in targeted tracer delivery. The CAM was studied during the growth phase of the chick embryo [16] while monitoring changes with serial imaging. We examined pectin-facilitated drug transport using a panel of six tracers with distinct physicochemical properties.

## 2. Methods

**Pectin**. A commercial source (Cargill, Minneapolis, MN, USA) was used for the pectin in this study. The plant source was citrus pectin. The proportion of galacturonic acid residues in the methyl ester form determined the degree of methoxylation. The high-methoxyl pectins used in these experiments had a degree of methoxylation in excess of 70%. The pectin powder was sterilely aliquoted and stored in a humidity-controlled environment at 25 °C.

**Pectin films**. The pectin powder (3% *w*/*w*) was dissolved at 25 °C by a staged procedure. The procedure was staged in a defined procedure to maximize solubility and minimize undissolved powder. No exogenous heat was used. Progressive dissolution was confirmed visually. The complete dissolution of the pectin was achieved using a high-shear 10,000 rpm rotor-stator mixer (L5M-A, Silverson, East Longmeadow, MA, USA). The solubilized pectin was cured in a variety of custom molds for further studies. The curing process, including the ambient humidity, depended upon water content and mold configuration. All curing processes were performed at 25 °C.

**Nanocellulose fibers (NCF).** Briefly, NCF was obtained from the Process Development Center at the University of Maine (Orono, ME, USA). The NCF powder was similarly aliquoted and stored in a sterile humidity-controlled environment. Similar to the pectin process, the NCF dissolution was obtained with progressive hydration in a step-wise procedure. Complete dissolution was obtained by a high-shear 10,000 rpm rotor-stator mixer (L5M-A, Silverson). Once the NCF powder was completely dissolved at 25 °C, the dissolved NCF was poured into polystyrene molds and cured for further studies.

**Contact angle**. To determine the wettability of the pectin surface, we measured contact angles. Contact angles provide insight into surface interaction sand the wettability of the pectin films. Contact angle measurements were based on a sessile 5 μL drop placed at multiple areas of the film surface. The droplets were placed carefully to ensure reproducibility. Custom Nikon SMZ stereomicroscope system oriented orthogonal to the plane of the film was used to image the droplet. A series of images was acquired using MetaMorph 7.10 acquisition software (Molecular Devices, Downingtown, PA, USA). The distance calibrated image was processed using standard filters of the MetaMorph software. The image contact angle was performed using a software goniometer; that is, a MetaMorph script accurately and rapidly captured the angle formed at the point where the liquid, pectin film, and gas (air) phases meet. Serial measurements were obtained to distinguish static and dynamic contact angles.

**Ex ovo cultures**. A standard ex ovo culture method was used for the chick chorioallantoic membrane [17]. The eggs were obtained from a commercial source. Egg viability was ensured with meticulous temperature and environmental control during shipping. Briefly, the eggs were maintained in a digital incubator (GimHae, Republic of Korea) with controlled conditions including 37.5 °C and 70% humidity. The eggs were kept in the incubator with automatic turning for 3 days. On embryonic development day (EDD) 3, the eggs were harvested. First, the eggs were sprayed with 70% ethanol and air-dried in a laminar flow hood. Second, the eggs were explanted into a 20 × 100 mm Petri dish (Falcon, BD Biosciences, San Jose, CA, USA). The ex vivo cultures were maintained in a meticulously maintained humidified 5% CO_2_ incubator at 37.5 °C.

**Active loading**. Active loading is a technique used to embed a high concentration of a substance within the carrier system. Our approach to the active loading of the tracers into the polymer films has been described elsewhere [10]. Briefly, we used compression with a 25 mm diameter acrylic disk to embed the tracer into pectin films using a 5 kg load cell (TA-XT plus; Stable Micro Systems, Godalming, UK). The disc was mounted to the crosshead over the center of a 25 mm pectin polymer placed on the fixture table. The 10 µL tracer droplet, at a constant concentration of 5 mg/mL, was placed on the film. The loading probe descended with tunable velocity, contact time, and compression force. Active loading of the low viscosity tracers was achieved with a probe velocity of 5 mm/s and compression force of 5 N.

**Linescan**. The linescan function in MetaMorph is a data acquisition technique that acquires a single line of data at a time. This approach provided a selective measure of tracer diffusion. To assess tracer diffusion from the original application site, we used the standard linescan function (MetaMorph) to measure color intensity values along a linear (35 mm × 6 mm) region of interest. Given the different tracer colors, RGB intensities were measured separately and the of the three wavelengths averaged to a single value.

**Color thresholding**. Color thresholding in image analysis is a technique for segmenting objects within an image based on object colors. To identify and segment the colored tracers, each 24-bit image was separated into intensity values for red, green, and blue color models. For each separated channel, a specific intensity value, based on linescan analysis, was set for the tracer threshold. In some cases, we converted the RGB image to another color space (HSV; Hue, Saturation, Value) to improve intensity separation. In each case, pre-processing was performed to facilitate diffusion-related tracer tracking.

**Morphometry analysis**. As previously described, imaging was obtained with care taken to minimize movements or temperature changes in the time series acquisition. Standard lighting and acquisition settings were used to obtain 12-megapixel images of the tracer patterns. Background subtraction and standard MetaMorph 7.10 software (Molecular Devices) acquisition and processing filters were applied. The 14-bit images were distance calibrated and processed through a color thresholding process. The images were measured using MetaMorph’s *Integrated Morphometry* application. irrespective of intensity variation, area was the total area of the number of pixels in the image.

**Statistical analysis**. The statistical analysis was based on measurements in replicate samples; in most cases, at least three different samples. The unpaired Student’s *t*-test for samples of unequal variances was used to calculate statistical significance. The data are expressed as mean ± one standard deviation. The significance level for the sample distribution was defined as *p* < 0.05.

## 3. Results

**Passive diffusion**. The color tracers were directly applied at identical concentrations and volumes to the CAM surface and imaged at hourly intervals for 5 h. The distance calibrated and intensity corrected images were assessed by planimetry (Figure 1). When applied directly to the CAM surface, the tracer profiles were relatively static. The tracers demonstrated some increase in surface area generally reflecting their solubility (Table 1) (Figure 2, circles). All tracers demonstrated limited diffusion into the albumin compartment (Figure 2, squares).

**Tracer-pectin interaction**. To evaluate the surface energy of the pectin-tracer interface, we evaluated the wetting behavior of the tracers on the pectin films using contact angles. The contact angle is the angle formed between the pectin surface and the tangent to the liquid surface at the 3-phase boundary (Figure 3A). All six of the tracers demonstrated acute angles consistent with a hydrophilic and wettable surface (Figure 3B). The wettability of the tracers was statistically similar with the exception of the crystal violet tracer (*p* < 0.01).

**Facilitated diffusion**. To facilitate tracer transport into the CAM, the tracers—at identical concentration and volumes—were actively loaded into pectin films [10]. Upon contact with the CAM surface, the pectin films became densely adherent. The methylene blue tracer demonstrated smoothing of the tracer margins within 1 h (Figure 4, gray arrow). Within 3 h, there was an apparent halo at the tracer margin (Figure 4, white arrows). The halo was demonstrable in all three cationic tracers: methylene blue, methyl green, and crystal violet. The halo was not observed in the three anionic tracers. Using linescan intensity measures and color thresholding, the surface area of the cationic tracers was significantly greater than the anionic tracers (Figure 5, *p* < 0.001). In addition to the halo surface area, diffusion into the CAM was confirmed by tracer detection in perivascular CAM lymphatics and color change in the embryo (Figure 6, circle). In contrast, identical concentration of tracer loaded into linear-branch NCF films demonstrated no significant CAM diffusion (Figure 6, triangles).

## 4. Discussion

The glycocalyx presents a formidable barrier to passive diffusion. Recent work suggests that the glycocalyceal barrier on the surface of visceral organs is 100- to 1000-fold thicker than previous estimates [8]. This dense physical barrier protects the underlying epithelium and impedes passive diffusion. Although the glycocalyx is an imposing barrier, the present study suggests that it is not impenetrable. In this report, we studied six color dyes commonly used as diffusion tracers in an ex ovo chorioallantoic membrane (CAM) model. When directly applied to the CAM surface, none of the tracers passively diffused into the membrane. Similarly, tracers embedded into linear-chain NCF films did not diffuse into the CAM membrane. In contrast, cationic tracers embedded into branched-chain pectin films were able to penetrate the CAM glycocalyx and its epithelial barrier. We conclude that tracer-loaded pectin films facilitated the diffusion of tracers across the glycocalyceal charge barrier.

The evidence for the charge barrier of the CAM glycocalyx is based on biochemical composition and interactions. In most tissues, the glycocalyx is rich in glycosaminoglycans like heparan sulfate, chondroitin sulfate, and hyaluronic acid [18,19]. These molecules have repeating disaccharide units that contain negatively charged groups such as sulfate and carboxylate ions. These components contribute to the overall negative charge of the glycocalyx. This negative charge is reflected by the interaction of the glycocalyx with ruthenium red [20]. A cationic molecule, the staining mechanism of ruthenium red, is based on the electrostatic interaction between the positively charged ruthenium ions and the negatively charged glycocalyceal components [21]. Ruthenium red has been used to effectively stain the glycocalyx in the CAM [22].

There is a subtle but relevant distinction between diffusion and permeation. Both diffusion and permeation describe processes related to the movement of molecules. In diffusion, the molecules move from a region of higher concentration to a region of lower concentration. The driving force is the concentration gradient of the molecules. Permeation also involves the movement of molecules along a concentration gradient. The distinction is that permeation involves the molecules passing through a permeable material; typically, permeation involves the absorption into the material, diffusion through the material, and subsequent release on the other side [23,24]. Here, we describe the movement of tracers across the glycocalyceal charge barrier as “diffusion” rather than “permeation” to reflect a more general process.

The rate and efficiency of facilitated diffusion is primarily determined by concentration gradient. In these experiments, we controlled concentration, surface area application, and temperature (25 °C). The molecular mass varied from 319 gm/mol (methylene blue) to 792 gm/mol (brilliant blue), but molecular size in this limited sample did not correlate with facilitated diffusion. The primary determinant of facilitated diffusion was the ionization of the tracer. The cationic tracers selectively diffused across the CAM with the aid of the hydrogel carrier. We anticipate that this observation can be tested with a broad range of charged functional molecules in future work.

We described the movement of the color tracers in these experiments as “facilitated diffusion”. In most biologic systems, facilitated diffusion describes the transit of molecules across the cell membrane facilitated by proteins embedded in the cell membranes [25]. These proteins can be either carrier proteins or channels that provide a mechanism for transmembrane transport [26]. In most cases, this transport mechanism is selective for a particular category of molecules. In our experiments, the pectin appeared to facilitate the transport of a category of color tracers across the glycocalyceal charge barrier. We speculate that the process observed in these experiments reflects facilitated transport. First, the process did not require energy, as tracers diffused along concentration gradients from a region of higher concentration to a region of lower concentration. Second, the process required a carrier molecule. Here, the carrier molecule was not a protein, but rather the heteropolysaccharide hydrogel pectin. We speculate that the entanglement of pectin chains with the glycocalyx facilitated transport of the cationic tracers across the glycocalyceal charge barrier.

Hydrogels used in targeted drug delivery can differ widely in polymer architecture and function [27,28]. In general, hydrogels are three-dimensional, hydrophilic polymer networks that can retain protein drugs and growth factors through either complicated chemical interactions [29] or simple physical embedding [30]. Physical embedding typically reflects drug retention in the polymers’ free volume. The relative size of the drug and the size of the polymer pores determine the kinetics of drug release. When the mesh size of the network is larger than the drug, the process is dominated by diffusion. Drug molecules smaller than mesh size diffuse freely. Diffusion is independent of mesh size. Larger drugs diffuse slower. Drugs sufficiently large relative to mesh size may require hydrogel erosion for drug release [31]. Here, we measured the appearance of the tracers in the three compartments: (1) on the surface of the CAM, (2) in the albumin compartment, and (3) within the chorioallantoic membrane. Although the rate of diffusion was not directly measured in this limited sample, our observations suggest that the diffusion of our limited number of tracers into the CAM occurred within hours and was independent of tracer size.

Because of their hydrophilic chemistry, hydrogels can absorb and retain significant amounts of water or therapeutic fluids which makes them appealing for various biomedical applications. A particularly intriguing hydrogel is the plant-derived heteropolysaccharide biopolymer called pectin. Pectin has a high content of partially esterified linear chains of (1,4)-alpha-D-galacturonic acid residues and a substantial free volume [32]. Pectin is bioabsorbable, biodegradable, and cyto-compatible [33]. Importantly, pectin has a unique mechanism of bioadhesion [11]. Pectin chains entangle with the surface glycocalyx of mammalian cells [34]. Entanglement results in strong bioadhesion to visceral organ surfaces. These studies suggests that pectin represents a practical approach to, and a therapeutic development path for, targeted drug delivery to the visceral organ surface.

In summary, we have used six widely available color tracers and the chick chorioallantoic membrane (CAM) model to investigate the phenomenon of facilitated diffusion across the glycocalyceal charge barrier. The structural heteropolysaccharide bioadhesive pectin was studied. Pectin is both a hydrogel with a substantial free volume as well as a strong bioadhesive to the visceral organ surfaces. The color tracers included brilliant blue, Congo red, crystal violet, indocyanine green, methylene blue, and methyl green. Direct application of the tracers to the CAM surface (no film) or embedding tracers into linear chain nanocellulose fiber films resulted in no significant diffusion into the CAM. In contrast, when the tracers were actively loaded into branched-chain pectin films, there was significant detectable diffusion of the tracers into the CAM. In this limited study, facilitated diffusion was observed in the three cationic tracers but was limited in the three anionic tracers. Diffusion appeared to be dependent on ionic charge, but independent of tracer size or molecular mass. We conclude that dye-loaded pectin films facilitated the efficient diffusion of color tracers across the glycocalyceal charge barrier and may provide a therapeutic path for drug delivery to the surface of visceral organs.

## Figures and Tables

**Figure 1 polymers-16-00004-f001:**
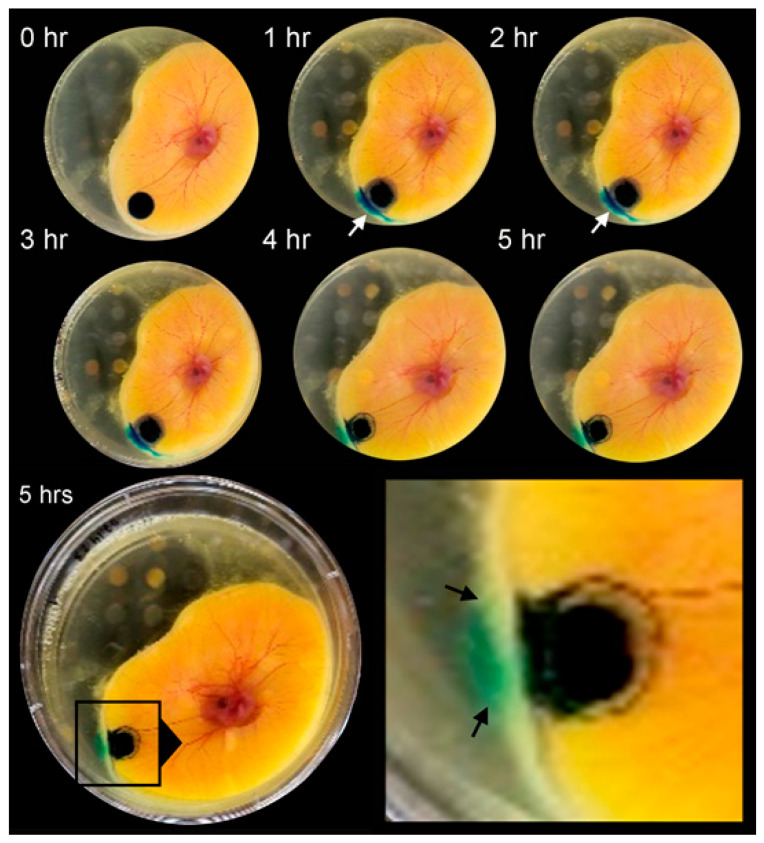
Passive diffusion of the tracer methylene blue on the ex ovo CAM. The methylene blue (10 µL at 5 mg/mL) was applied directly to the CAM and observed for the 5 h study period. By 3–4 h, some tracer was detected in the contiguous albumin compartment of the ex ovo CAM (arrows). There was no detectable tracer in the CAM or the embryo.

**Figure 2 polymers-16-00004-f002:**
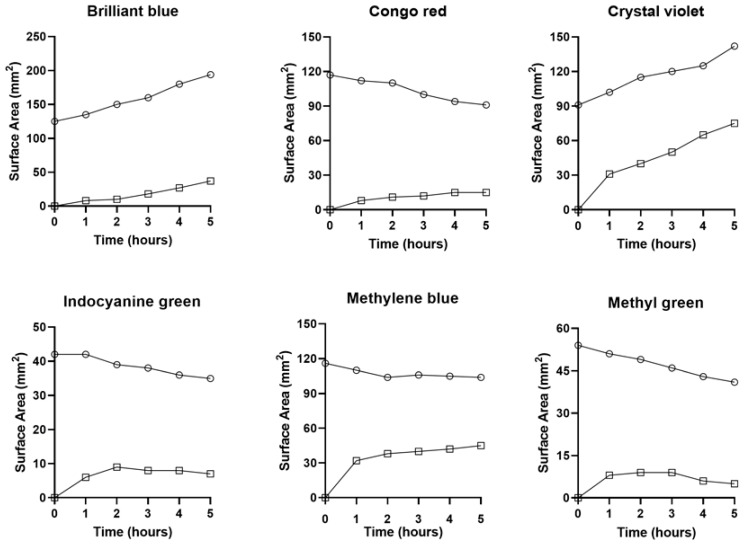
Planimetry of the 6 tracers after direct application to the ex ovo CAM surface. Each tracer was added at a volume of 10 µL and a concentration of 5 mg/mL. The site of application on the CAM was remote from the embryo. The surface area of the application site (circles) and the collateral diffusion area (squares) was calculated at each time point. The ex ovo CAM cultures were maintained at 37 °C with images obtained orthogonal to the CAM surface at hourly intervals from 0 to 5 h. The images were distance calibrated and processed by standard MetaMorph filters. Baseline cardiac function ensured CAM viability; 3 to 5 replicates were obtained for each time point. A representative time course for each tracer is shown.

**Figure 3 polymers-16-00004-f003:**
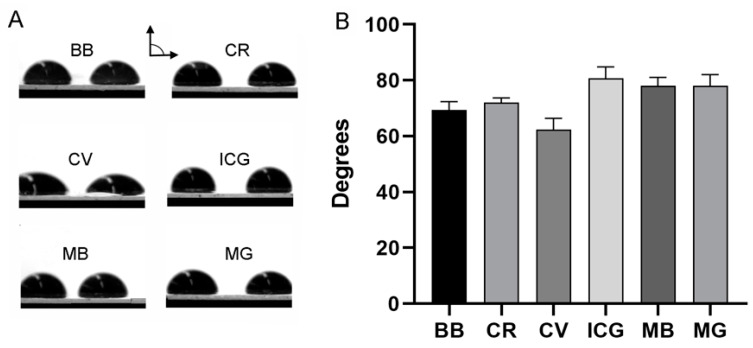
Surface interactions of the color tracers with the pectin film. (**A**) A sessile 5 μL drop of color tracer was placed on each film. The contact angle, the angle formed between the pectin surface and the tangent to the liquid surface at the 3-phase boundary, was measured using MetaMorph’s angle measurement application. (**B**) The acute angles obtained with all six tracers was consistent with a hydrophilic and wettable surface. The only statistically different tracer was crystal violet (*p* < 0.01). BB, brilliant blue; CR, congo red; CV, crystal violet; ICG, indocyanine green; MB, methylene blue; MG, methyl green. Error bars reflect 1 S.D. Replicate droplets are shown.

**Figure 4 polymers-16-00004-f004:**
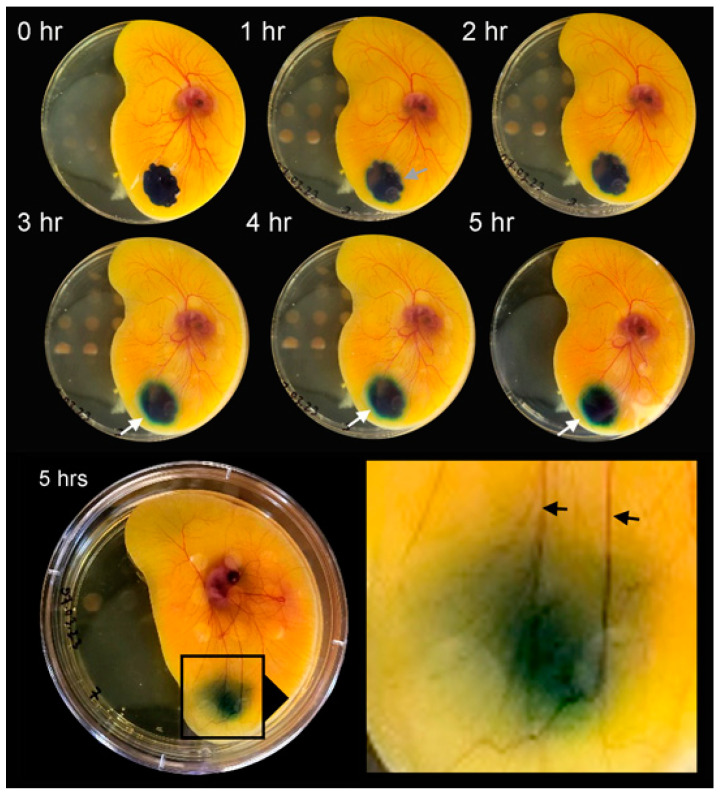
Facilitated diffusion of the tracer methylene blue on the ex ovo CAM. The methylene blue (10 µL at 5 mg/mL) was embedded in a pectin film. The film was applied to the CAM surface and observed for the 5 h study period. By 1–2 h, the lobulated margins of the tracer were blunted (gray arrow). By 3–4 h, a tracer halo was observed (white arrows). By 5 h, tracer was also visible in the perivascular lymphatics (black arrows) and detectable color change in the embryo.

**Figure 5 polymers-16-00004-f005:**
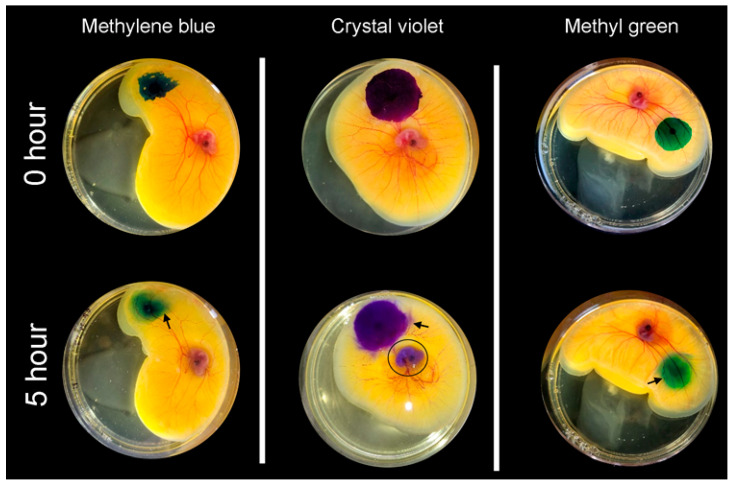
Facilitated diffusion of the cationic tracers on the ex ovo CAM. The tracers (10 µL at 5 mg/mL) were actively loaded into pectin films. The films were applied to the CAM surface and observed for the 5 h study period. In all three tracers, a halo was observed in the CAM (arrows). Corresponding color was also detectable in the embryo; the color change was most apparent with the crystal violet dye (circle).

**Figure 6 polymers-16-00004-f006:**
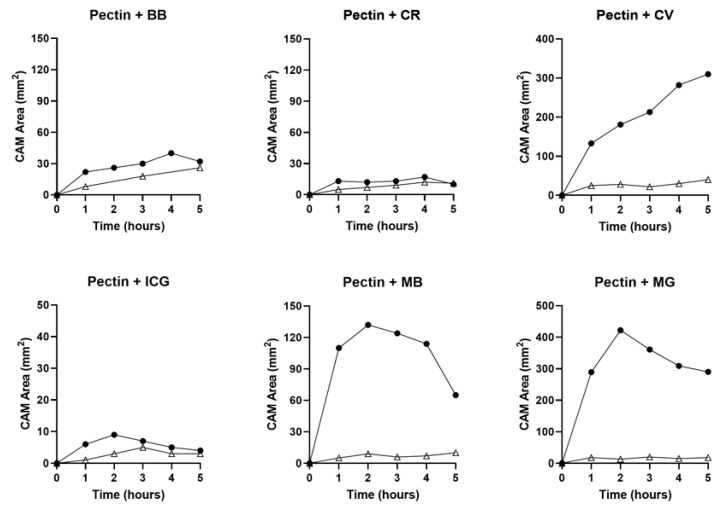
Planimetry of the six tracers embedded into carrier hydrogels and applied to the ex ovo CAM surface. The area of the tracer diffusion into the CAM is shown. The branched-chain pectin films (circles) were compared to the linear chain NCF films (triangles). Each tracer was embedded into pectin or NCF films at a volume of 10 µL and a concentration of 5 mg/mL. The site of application of the film was on the CAM but remote from the embryo. The ex ovo CAM cultures were maintained at 37 °C (5% CO_2_) with imaging obtained orthogonal to the CAM surface at hourly intervals for 5 h. After distance calibration, the surface area of the pectin site (circles) and the NCF site (triangles) was calculated at each time point; 3 to 5 replicates were obtained for each time point. The CAM data was excluded with a loss of CAM viability at any point during the study period. A representative time course for each tracer is shown.

**Table 1 polymers-16-00004-t001:** Physicochemical Properties of Color tracers.

Name	Formula	MW ^1^	Color	Solubility ^2^	Ionization
Brilliant blue	C_45_H_44_N_3_NaO_7_S_2_	792	Blue	30	Anionic
Congo Red	C_32_H_22_N_6_Na_2_O_6_S_2_	696	Red	33	Anionic
Crystal violet	C_25_H_30_ClN_3_	408	Violet	50	Cationic
Indocyanine Green	C_43_H_47_N_2_NaO_6_S_2_	774	Green	43	Anionic
Methylene Blue	C_16_H_18_ClN_3_S	319	Blue	1	Cationic
Methyl Green	C_32_H_37_N_4_Cl	364	Green	1	Cationic

^1^ Molecular weight is expressed as molar mass (g/mol). ^2^ Solubility in water at room temperature in gm/mL.

## Data Availability

Data available with appropriate request.

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
