# Peer review of "Facilitated Transport across Glycocalyceal Barriers in the Chick Chorioallantoic Membrane"

_polymers, 2023, doi:10.3390/polym16010004_

Round 1

Reviewer 1 Report

Comments and Suggestions for Authors

1. The numerical value and mean of main finding should be included in Abstract with the additional information of color name. 

2. In Introduction part, more review on CAM role on active compounds permeation studied with results of previous studied and citation should be added. While the angiogenesis check with CAM how could it relate with your experiment.

3. For Introduction, the rational indication of using dyes instead of model drug should be addressed with the physicochemical characteristics of different tracer colors employed in this research.

4. Please correct the chemical formula in Table 1 in term of scientific style. 

5. S.D. or y error bar should add in line graphs of Fig. 2, 6.

6. Practically, the viscosity of solution such as dye solution affects the parameters of contact angle and penetration; thus, the dyes solution should be measured their viscosity. In addition, charge of dye might influence on viscosity.

7. Why does CAM is better than cornea or retina of cow eye for testing penetration or permeation with glycocalyceal barrier?

8.  "When directly applied to the CAM surface, none of the tracers passively  diffused into the membrane. Similarly, tracers embedded into linear-chain NCF films did not diffuse into the CAM membrane." How do the authors make sure that the diffusion related with permeation? 

9. How did the authors control the flow of dye solution on curved surface of CAM. 

Author Response

Reviewer #1

  1. The numerical value and mean of main finding should be included in Abstract with the additional information of color name. 

A:  We have added the tracer names.  As noted below, the biologic variability precluded an easy numeric analysis, however, we summarized the relevant data in the Abstract.

  1. In Introduction part, more review on CAM role on active compounds permeation studied with results of previous studied and citation should be added. While the angiogenesis check with CAM how could it relate with your experiment.

A:  We have added a paragraph on permeation versus diffusion in the Discussion on page 9.  We have also clarified the importance of serial imaging—easy in the CAM and very difficult in other tissues (e.g cornea).

  1. For Introduction, the rational indication of using dyes instead of model drug should be addressed with the physicochemical characteristics of different tracer colors employed in this research.

A:  The use of color tracers facilitate visual examination in the CAM model.  This is highlighted in the Introduction on page 3.

  1. Please correct the chemical formula in Table 1 in term of scientific style. 

A: Done.

  1. S.D. or y error bar should add in line graphs of Fig. 2, 6.

A: Our statistical consultants at the Harvard School of Public Health have analyzed our data set.  The biological variability of the CAM model precludes traditional analysis of variance with SD and error bars.  They suggested using the median values to provide representative data in Figures 2 and 6. The data in these figures represents the median values as noted in the revised manuscript.

  1. Practically, the viscosity of solution such as dye solution affects the parameters of contact angle and penetration; thus, the dyes solution should be measured their viscosity. In addition, charge of dye might influence on viscosity.

A: We agree. Viscosity, however, will reflect the physiologic diluent.  The constant color tracer concentrations were chosen to provide rapid dilution in the CAM fluid.  We suspect this methodology limited the viscosity impact.

  1. Why does CAM is better than cornea or retina of cow eye for testing penetration or permeation with glycocalyceal barrier?

The CAM provides for direct visualization and serial imaging of tracer diffusion. Other models, such as the cow cornea, do not.

  1. "When directly applied to the CAM surface, none of the tracers passively  diffused into the membrane. Similarly, tracers embedded into linear-chain NCF films did not diffuse into the CAM membrane." How do the authors make sure that the diffusion related with permeation? 

We assume that tracer movement, from higher concentration to lower concentration , was due to the natural namdom motion of the tracers.  The primary driving force is concentration gradient. In contrast permeation is a process involving molecules passing through a permeable material.  We have specifically addressed this issue in the Discussion on page 8 and 9.

  1. How did the authors control the flow of dye solution on curved surface of CAM. 

We routinely use laser topography mapping to identify flat regions of the CAM.  Topographic variability between samples was limited.

Reviewer 2 Report

Comments and Suggestions for Authors

1. The authors need to present the interaction scheme of pectin  with anionic and cationic tracers. 

2. What is the reason that tracer-loaded pectin films differently promote diffusion of tracers across the glycocalycium charge barrier depending on the tracer charge?

3. On page 9 (line 208) the authors write: "The molecular mass varied from 319 to 792 gm/mol" What does it mean?

Author Response

Reviewer #2

  1. The authors need to present the interaction scheme of pectin with anionic and cationic tracers. 

A: Our attempts at schematic were unsatisfactory.  Instead we have added two paragraphs in the Discussion  on page 8 and 9 to address this issue.

  1. What is the reason that tracer-loaded pectin films differently promote diffusion of tracers across the glycocalyceal charge barrier depending on the tracer charge?

A:  We added one paragraph in the Discussion on page 9 to clarify this observation.

  1. On page 9 (line 208) the authors write: "The molecular mass varied from 319 to 792 gm/mol" What does it mean?

A:  We clarified the sentence: “The molecular mass varied from 319 gm/mol (methylene blue) to 792 gm/mol (brilliant blue).”